# GENERALIZATION ERROR ANALYSIS OF DEEP PHYSICAL MODELS WITH LATENT VARIABLES TRAINED ON TRAJECTORY DATA

## ABSTRACT

In this paper, we investigate the generalization error of deep physical models with latent variables. Deep physical models, such as Hamiltonian Neural Networks, are neural network models for learning equations of motion from observational data of physical phenomena and have attracted much attention in recent years. In particular, in such cases, the data are not completely random, but rather given as random trajectories. We provide an error bound for the case where the training data are given in such a way. Our results show that it is important to collect data from many trajectories, rather than simply collecting a large number of data, to improve generalization performance. In addition, an important application of the combination of deep physics models with latent variables is the interpolation of images from videos while preserving the laws of physics, such as the energy conservation law. However, when the frame interval of the video is large, it can be difficult to preserve the laws of physics. In this paper, we show that it is possible to interpolate the images from videos so that the laws of physics are preserved, provided that the generalization error is sufficiently small.

## 1 INTRODUCTION

Recently, deep learning methods have attracted much attention for learning the equations of motion from observational data of physical phenomena. In this paper, we investigate the generalization error of such models when the models are trained with the random trajectory data. In addition, an important application of deep physical models is the interpolation of images from videos while preserving the laws of physics, such as the energy conservation law. As an application of the error analysis, we also show that it is possible to interpolate the images from videos so that the laws of physics are preserved, provided that the generalization error is sufficiently small.

Although neural networks have been applied not only to modeling physical phenomena but also to other important tasks that include solving partial differential equations such as physics-informed neural networks (Raissi et al., 2019) and neural operators (Li et al., 2020; Kovachki et al., 2023; Lu et al., 2021), in this paper we focus on the neural network models for modeling and call them deep physics models for simplicity.

Typically behind the physical phenomena is analytical mechanics, which is the theory of classical mechanics that allows the modeling of complex and non-linear dynamical systems. There are two theories of analytical mechanics, the Lagrange and the Hamilton mechanics. Most of the deep physical models are based on the Hamilton equation, which is the equation of motion of Hamiltonian mechanics:

$$\frac{\mathrm{d}u}{\mathrm{d}t} = S\frac{\partial H}{\partial u}, \tag{1}$$

where $u : t \in \mathbb{R} \mapsto u(t) \in \mathbb{R}^N$, $H : u \in \mathbb{R}^N \mapsto H(u) \in \mathbb{R}$. Typically, $u$ is given as $(q, p)$, where $q$ represents the state variables and $p$ the generalized momenta. $S$ is a skew-symmetric matrix and $H$ is a real-valued function of $u$, which represents the total energy of the system. The most representative model for extracting the equation of motion is the Hamiltonian neural networks (HNNs, Greydanus

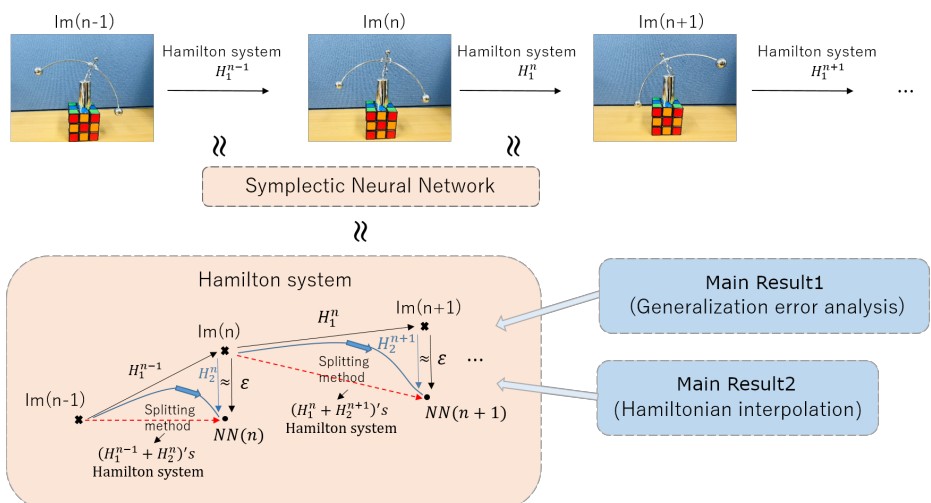

Figure 1: Overview of the research. Interpolation is performed on a sequence of images representing a physical phenomenon while respecting the laws of physics. If the modeling error of the learned model is small enough, the difference between the true Hamiltonian and the model can be approximated by a Hamiltonian system with very high accuracy. This can be used to construct a Hamiltonian system that approximates the model.

et al. (2019))

$$\frac{\mathrm{d}u}{\mathrm{d}t} = S\frac{\partial H_{\mathrm{NN}}}{\partial u}, \qquad (2)$$

where $H_{\mathrm{NN}}$ is a neural network, and various extensions of this model have been proposed. Based on the theory of physics, these models are designed so that the laws of physics, such as the energy conservation law, are maintained.

On the other hand, theoretical analysis is limited and, in particular, no error analysis has been performed in the practical situations where the models are trained with trajectory data. A generalization error bound of HNNs is investigated in Chen et al. (2021a), in which the training data are assumed to be given completely at random. However, when these models are used, the data are not completely random, but rather given as random trajectories. In addition, a typical application of such models is the interpolation of the images from videos while preserving the laws of physics. For this application, the models are combined with the autoencoder; however, to the best of the authors' knowledge, there are no theoretical analyses of the performance of such models. In particular, as explained in Section 3.2, it is not obvious whether the trained models in fact preserve the laws of physics or not.

In this study, first, we present a generalization error bound for the deep physical models including HNNs with and without the autoencoder. Theoretical analysis of generalization error is typically based on statistical learning theory, in which it is usually assumed that the data are random samples from a certain probability distribution, thereby evaluating the difference between the loss function for the training data and the expected value of the loss function. However, as explained above, when neural networks for modeling the equation of motion are used, the training data are given as trajectory data. More precisely, the initial values of the data are given randomly, but the subsequent data are generated according to the laws of physics. Therefore, only the initial value of each trajectory can be given at random; as far as we know, this problem setting has not been considered yet in the existing theoretical analyses for such models.

Second, as an application of the generalization error analysis, we prove that the so-called Hamiltonian interpolation is possible by combining the autoencoder and symplectic neural network models. As is demonstrated in the original paper of HNN (Greydanus et al., 2019), an important application of these models is learning the equation of motion from images by combining the autoencoder, thereby interpolating the images while preserving the laws of physics. For example, when creating an animation, if several images with large time intervals can be prepared and interpolated between them, the animation can be created efficiently. To avoid unnaturalness, it is desirable to interpolate

Table 1: Comparison with other theoretical results for deep physical models.

|  | SympNet (Jin et al., 2022) | GFINNs (Zhang et al., 2022) | KAM (Chen et al., 2021a) | Our Results |
|---|---|---|---|---|
| Universal Approximation Theorem | yes | yes | yes |  |
| Generalization Error Analysis |  |  | yes | yes |
| Learning from Pixels |  |  |  | yes |
| Random Trajectory Data |  |  |  | yes |

the images while respecting the laws of physics. In such an application, whether the interpolated images follow a Hamiltonian equation is very important. Meanwhile, the problem of determining if a given function is obtained from a Hamiltonian equation or not has been studied in the fields of symplectic geometry and numerical analysis. By combining the generalization error bound and these theories, particularly the so-called splitting method, we will show that if the number of data is sufficient, with a high probability the interpolation is possible while preserving the laws of physics.

The main contributions of this paper are as follows (see also Figure 1 and Table 1.)

- We establish a theoretical framework for analyzing the generalization error when the initial values of each trajectory in the data can be given at random. As far as we know, there is no theoretical analysis performed in this setting. It is shown that the generalization error depends on the number of trajectories, not on the total number of the data.

- By combining this generalization error analysis with the theory of symplectic geometry and the error analysis of a numerical integrator called the splitting method, we theoretically prove that with a high probability, there exists a Hamiltonian system that interpolates the given image sequences with large time intervals. This result presents a new direction in the theoretical analysis of deep physical models in the sense that it is a combined study of three fields: statistical learning theory, symplectic geometry, and numerical analysis.

## 2    RELATED WORK

Since ordinary differential equations describe general dynamical systems, Neural Ordinary Differential Equation (NODE, Chen et al. (2018)) is a general model for modeling ordinary differential equations. However, due to the generality of neural networks, this method does not necessarily work well for modeling physical phenomena. Most of the existing deep learning models for physics are based on analytical mechanics. For example, the Lagrangian Neural Networks (LNN, Cranmer et al. (2020)) learns the Lagrangian from data and derives the equation of motion as the Euler–Lagrange equation. Hamiltonian Neural Networks (HNN, Greydanus et al. (2019)) is a neural network model for Hamiltonian systems, in which the Hamiltonian, the energy function, is modeled using neural networks. There are many extensions of HNNs, for example, symplectic ODE-Net (Zhong et al., 2019), Variational Integrator Networks (Saemundsson et al., 2019), DGNet(Matsubara et al., 2019), neural symplectic forms (Chen et al., 2021b), Poisson Neural Networks (Jin et al., 2022).

At the same time, these deep physical models are applied to derive equations of motion from animations. The currently available method includes the Hamiltonian Generative Network (Toth et al., 2019), which is capable of learning Hamiltonian dynamics from high-dimensional observations (e.g. images.) Similarly, KeyCLD(Daems et al., 2022) is a framework for learning Lagrangian dynamics from images. Khan & Storkey (2021) focused on a deep generative model utilizing Hamiltonian Latent Operators to reliably disentangle content and motion information in image sequences. Mason et al. (2022) presented a physics-informed neural network model to estimate and predict 3D rotational dynamics from image sequences with a multi-stage prediction pipeline. Hofherr et al. (2022) proposed to combine neural implicit representations for appearance modeling with neural ordinary differential equations for modeling physical phenomena to obtain a dynamic scene representation from visual observations. Jatavallabhula et al. (2021) proposed gradsim which addressed visuomotor control tasks without relying on state-based supervision.

From the perspective of theoretical research, Jin et al. (2020) proposed SympNet along with the universal approximation theorem for the proposed model. HNNs were shown to have the universal approximation property in Chen et al. (2021a) along with an application to the KAM theory. Zhang et al. (2022) proposed the GENERIC formalism informed neural networks that obey the degeneracy conditions of the GENERIC formalism, and proved the universal approximation theorem for their models.

As far as we know, no theoretical analysis of deep learning models trained with trajectory data has been performed. In addition, the Hamiltonian interpolation using these models has not been investigated.

## 3 BRIEF INTRODUCTION OF DEEP PHYSICAL MODELS WITH LATENT VARIABLES AND SUMMARY OF EXISTING THEORIES

### 3.1 DEEP PHYSICAL MODEL BASED ON HAMILTONIAN MECHANICS WITH LATENT VARIABLES

In this paper, we consider the combination of deep physical models based on Hamiltonian mechanics and the autoencoder for modeling the equation of motion from images. We call these models deep physical models with latent variables. Note that such models include the deep physical models alone because the encoder and the decoder of the autoencoder can be the identity map.

In deep physical models with latent variables the features extracted by the autoencoder are used as the state variables of the deep physical models. More precisely, we consider typical models that have the architecture shown in Figure 2. To train these models, first, the features of the images are extracted by the encoder. Then the extracted features are sent directly to the decoder to compute the loss function between the output results and the original data. On the other hand, the extracted features are also input into the deep physical model to compute the loss function between the prediction by the deep physical model and the features of the next image computed by the autoencoder.

Obviously, as the deep physical model, arbitrary neural network models for physics can be employed. In this paper, we perform a generalization error analysis and a theoretical analysis on the Hamiltonian interpolation. In the generalization error analysis, we do not place any restriction on the type of the neural network models; for example, the models for energy-dissipative phenomena (e.g., pendulum with friction) can be used. For the latter part, we only consider Hamiltonian systems and assume that deep physical models define a symplectic map.

### 3.2 SYMPLECTIC INTEGRATORS

As is well known in the field of numerical analysis, symplectic numerical integrators are useful for numerically computing solutions to Hamiltonian equations. Symplectic integrators are designed so that the time evolution map given by the integrators is a symplectic map. For such an integrator, it is known that there exists a so-called shadow Hamiltonian, which defines a Hamiltonian system that approximates the symplectic map corresponding to the integrator. Motivated by this fact, several methods have been proposed to learn the discrete-time Hamilton equation in such a way that the neural network model becomes a symplectic map when learning a time-stepping map.

However, a symplectic map is guaranteed to be related to the Hamilton equation if it is close to the identity map (Benettin & Giorgilli, 1994; Reich, 1999). This is guaranteed for symplectic integrators when the time-step sizes are small enough; however, particularly in the case where the time-step sizes are not so small, it is not necessarily guaranteed that the symplectic map corresponds to the Hamilton equation.

Meanwhile, in the case of learning from video images, the time increments are not always small, and this assumption is generally not satisfied. Therefore, even if the model is constructed to be a symplectic map, it may not be possible to interpolate the images using the Hamilton equation.

### 3.3 Hamiltonian interpolation of symplectic mappings

As explained in the above, although any trajectory of a Hamilton equation defines a symplectic map, a symplectic map does not necessarily correspond to a Hamilton equation (Benettin & Giorgilli, 1994). It is known that if a symplectic map is close to the identity map, then there exists a Hamilton equation that approximates this map to a very high degree of accuracy.

**Theorem 3.1 (Benettin & Giorgilli (1994) )** *For any symplectic mapping $\phi_\varepsilon$ that is analytic and $\varepsilon$-close to the identity, there exists an analytic autonomous Hamiltonian system, $H_\varepsilon$ such that the difference between the time-one mapping and $\phi_\varepsilon$ is $O(\varepsilon \exp(-1/\varepsilon))$.*

This theorem has been applied by Reich to evaluate the long-term performance of symplectic numerical integrators (Reich, 1999). In Reich (1999), it is shown that the numerical solutions by the symplectic numerical integrators can be practically regarded as a solution to a Hamilton equation. The Hamiltonian for this Hamilton equation is often called the shadow Hamiltonian. In fact, the symplectic numerical integrators define a symplectic map that is close to the identity map if the time-step size is sufficiently small. Thus, the above theorem is applicable, and the existence of a Hamiltonian system that approximates the symplectic map with an exponentially small error is guaranteed. However, the symplectic map learned by the neural network does not have a parameter corresponding to the time-step size. Therefore, the map may not be sufficiently close to the identity map, and even if the neural network model is symplectic, the existence of the corresponding Hamiltonian system is not obvious.

### 3.4 Splitting method

In the following section, we show that the Hamiltonian interpolation is certainly possible under certain conditions. The proof of this statement relies on the splitting method, which is a technique of numerical analysis. The splitting method is a numerical method for ordinary differential equations, particularly for Hamiltonian systems. Suppose that the Hamiltonian $H$ of a Hamiltonian system is represented by the sum of two Hamiltonians $H_1$ and $H_2$:

$$H(u) = H_1(u) + H_2(u).$$

Then, the time-1 mapping of the Hamiltonian system with respect to $H$ can be approximated by using the time-1 mapping of the Hamiltonian system with respect to $H_1$ and $H_2$. This method is known as the splitting method. As is well known, the difference between these two mappings is shown to be $O(|H_1||H_2|)$ by the Campbell–Baker–Hausdorff lemma (Hairer et al., 2006).

### 3.5 Tools for generalization error bounds

In statistical learning theory, generalization error bounds are obtained by using the Rademacher complexities or the covering number. See, e.g., Bousquet et al. (2004); Giné & Nickl (2016); Shalev-Shwartz & Ben-David (2014); Steinwart & Christmann (2008) for details.

**Definition 3.2** *For a set $V \subset \mathbb{R}^n$,*

$$\mathcal{R}_n(V) := \frac{1}{n}\mathbb{E}_{\sigma \sim \{-1,1\}^n} \sup_{v \in V} \sum_{i=1}^n \sigma_i v_i$$

*is called the Rademacher complexity of $V$.*

**Lemma 3.3** *Let $X$, $Y$ be arbitrary spaces, $\mathcal{F} \subset \{f : X \to Y\}$ be a hypotheses class, and $L : Y \times Y \to [0, c]$ be a loss function. For a given data set $(x_i, y_i) \in X \times Y$ $(i = 1, \ldots, m)$, let $\mathcal{G}$ be defined by $\{(x_i, y_i) \in X \times Y \mapsto L(y_i, h(x_i)) \in \mathbb{R} \mid h \in \mathcal{F}, i = 1, \ldots, m\}$. Then for any $\delta > 0$ and any probability measure $P$, we have with a probability at least $(1 - \delta)$ with respect to the repeated sampling of $P^m$-distributed training data*

$$E[L(Y, h(X))] \leq \frac{1}{m}\sum_{i=1}^m L(y_i, h(x_i)) + 2\mathcal{R}_m(\mathcal{G}) + 3c\sqrt{\frac{2\ln\frac{4}{\delta}}{m}}$$

*for all $h \in \mathcal{F}$.*

An estimate of the Rademacher complexity can be obtained by using the covering number.

**Definition 3.4** *Let $V, V'$ be subsets of $\mathbb{R}^n$. $V$ is $r$-covered by $V'$ with respect to the metric function defined by the norm $\| \cdot \|$ if for all $v \in V$ there exists a $v' \in V'$ such that $\|v - v'\| < r$. The covering number $N(r, V, \| \cdot \|)$ of $V$ is the minimum number of the elements of a set that $r$-covers $V$.*

We also denote $N(r, V, \| \cdot \|)$ by $N(r, V)$ when the norm is clear from the context.

**Lemma 3.5** *For $V \subset \mathbb{R}^m$ it holds that*

$$R(V) \leq \frac{6c}{m} \sum_{k=1}^{\infty} 2^{-k} \sqrt{\log(N(c2^{-k}, V))},$$

*where $c$ is the radius of the set $V$: $c = \min_{v_0 \in V} \max_{v \in V} \|v - v_0\|$. For example, if $\sqrt{\log N(c2^{-k}, V)} \leq \alpha + k\beta$ for some $\alpha$ and $\beta$, then $\mathcal{R}_n(V) \leq \frac{6c}{n}(\alpha + 2\beta)$.*

Thus, if the covering number is estimated for neural networks models, the bound on the generalization error is obtained. To estimate the covering number, the following lemmas are often used.

**Lemma 3.6** *Let $B_{\rho_B}$ be a ball with the radius $\rho_B$ in $\mathbb{R}^m$. Then it holds that $N(r, B_{\rho_B}, \| \cdot \|_2) \leq \left(\frac{2\rho_B}{r} + 1\right)^m$.*

**Lemma 3.7** *For any $V \subset \mathbb{R}^m$, $c > 0$, $v \in \mathbb{R}^m$, $r > 0$, it holds that $N(cr, \{cv + v_0 \mid v \in V\}) \leq N(r, V)$.*

**Lemma 3.8** *Suppose that functions $\phi_i : \mathbb{R} \to \mathbb{R}$, $i = 1, 2, \ldots, m$ are $\rho$-Lipschitz continuous. For $v \in \mathbb{R}^m$, let $\vec{\phi}(v)$ be defined by*

$$\vec{\phi}(v) := (\phi_1(v_1), \ldots, \phi_m(v_m)), \quad \vec{\phi} \circ V := \{\vec{\phi}(v) \mid v \in V\}.$$

*Then it holds that $N(\rho r, \vec{\phi} \circ V) \leq N(r, V)$, or equivalently, $N(r, \vec{\phi} \circ V) \leq N(\frac{r}{\rho}, V)$.*

## 4 MAIN RESULTS

In the following sections, we perform a generalization error analysis of a model with the structure shown in Figure 2, which combines the autoencoder and a deep physics neural network model, particularly in the case where the model is trained with trajectory data. Note that this model includes the deep physical model alone because the encoder and the decoder in the autoencoder can be the identity map. Hence our generalization error bound can be applied to the deep physical models without the autoencoder.

In addition, if the physics model defines a symplectic map, we also show that this model can be used for interpolating the images while preserving laws of physics even when the time intervals are quite large. By combining the results of the generalization error analysis, Hamiltonian interpolation of near-identity symplectic maps and the error analysis of the splitting method for Hamiltonian systems, we show that the model with this structure can interpolate images while satisfying a Hamiltonian equation even when the time interval is large.

### 4.1 GENERALIZATION ERROR ANALYSIS OF THE MODELS TRAINED WITH TRAJECTORY DATA

We first show a generalization error bound. We suppose that the model consists of three dense neural networks:

$$f_{\mathrm{NN}} : \mathbb{R}^{n_d} \to \mathbb{R}^{n_h}, \ g_{\mathrm{NN}} : \mathbb{R}^{n_h} \to \mathbb{R}^{n_d}, \ h_{\mathrm{NN}} : \mathbb{R}^{n_h} \to \mathbb{R}^{n_h}.$$

$f_{\mathrm{NN}}$ and $g_{\mathrm{NN}}$ define an autoencoder: $f_{\mathrm{NN}} \circ g_{\mathrm{NN}} \simeq \mathrm{Id}$. $h_{\mathrm{NN}}$ is assumed to be a symplectic map if it is used for Hamiltonian interpolation. As for the neural networks, for simplicity, we consider simple networks with one hidden layer:

$$f_{\mathrm{NN}}(x) = \sigma_{\mathrm{enc}}(A_{\mathrm{enc}}x + b_{\mathrm{enc}}), \ g_{\mathrm{NN}}(x) = \sigma_{\mathrm{dec}}(A_{\mathrm{dec}}x + b_{\mathrm{dec}}), \ h_{\mathrm{NN}}(x) = \sigma_{\mathrm{symp}}(A_{\mathrm{symp}}x + b_{\mathrm{symp}}).$$

The extension to deeper networks and/or deep physics neural networks like HNNs is straightforward.

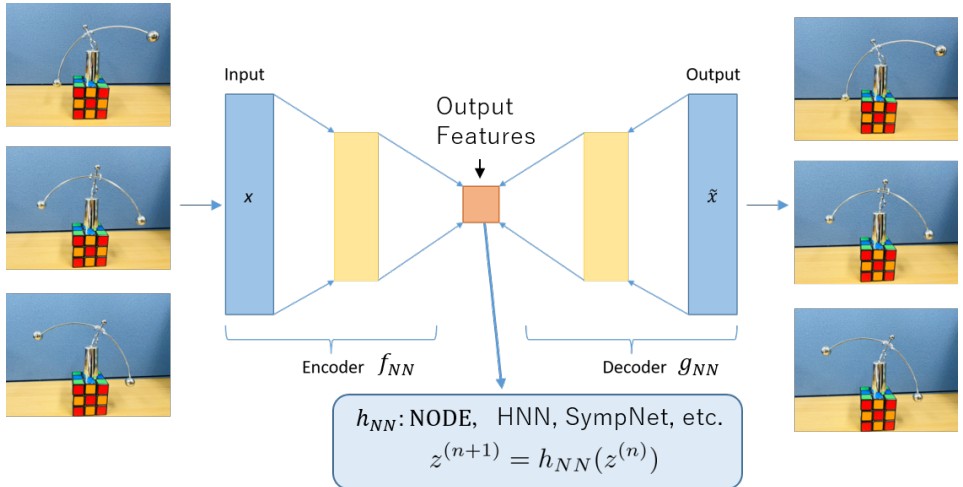

Figure 2: Architecture of the model.

**Assumption 4.1** *The data set is given as* $X = \{x_j^{(1)}, \ldots, x_j^{(m_{\mathrm{step}})} \mid j = 1, \ldots, m_{\mathrm{orbit}}\} \subset \mathbb{R}^{n_d \times m_{\mathrm{step}} \times m_{\mathrm{orbit}}}$, *where* $x_j^{(1)}$'s *are i.i.d. samples from an unknown distribution* $D$. *For each* $j$, $x_j^{(1)}, x_j^{(2)}, \ldots, x_j^{(m_{\mathrm{step}})}$ *are samples from an orbit starting from* $x_j^{(1)}$. $m_{\mathrm{orbit}}$ *is the number of the orbits. More precisely, we assume that there exists a* $\rho_\psi$-*Lipschitz continuous map* $\psi : \mathbb{R}^{n_d} \to \mathbb{R}^{n_d}$ *such that for each* $j$ *and* $n$, $x_j^{(n+1)} = \psi(x_j^{(n)})$. *We also assume that* $x_j^{(n)}$'s *are uniformly bounded so that there exists a compact set* $B \subset \mathbb{R}^{n_d}$ *such that* $x_j^{(n)} \in B$ *for any* $j$ *and* $n$.

We suppose that the model is trained by minimizing the following loss functions:

$$\text{minimize } \frac{1}{m_{\mathrm{orbit}}} \sum_{j=1}^{m_{\mathrm{orbit}}} \mathcal{L}(\{x_j^{(1)}, \ldots, x_j^{(m_{\mathrm{step}})}\}),$$

$$\mathcal{L} = \mathcal{L}_1 + \mathcal{L}_2, \quad \mathcal{L}_1(\{x_j^{(1)}, \ldots, x_j^{(m_{\mathrm{step}})}\}) = \frac{1}{m_{\mathrm{step}}} \sum_n l_1(g_{\mathrm{NN}}(f_{\mathrm{NN}}(x^{(n)})) - x^{(n+1)}),$$

$$\mathcal{L}_2(\{x_j^{(1)}, \ldots, x_j^{(m_{\mathrm{step}})}\}) = \frac{1}{m_{\mathrm{step}}} \sum_n l_2(h_{\mathrm{NN}}(f_{\mathrm{NN}}(x^{(n)})) - f_{\mathrm{NN}}(x^{(n+1)}))$$

where $l_1$ and $l_2$ are loss functions.

**Assumption 4.2** *We assume that there exist a latent variable* $z(t) \in \mathbb{R}^{n_h}$, *functions* $f : \mathbb{R}^{n_d} \to \mathbb{R}^{n_h}$ *and* $g : \mathbb{R}^{n_h} \to \mathbb{R}^{n_d}$ *that reconstruct the data* $x_j^{(n)}$'s *from* $z$. $n_h$ *is the dimension of the latent space. We also assume that* $z$ *satisfies a Hamiltonian equation, and for its existence, we assume that* $n_h$ *is even. More precisely, we assume that for any* $x_j^{(n)} \in X$, *it holds that*

$$x_j^{(n)} = g(f(x_j^{(n)})), \ f(x_j^{(n)}) = z((n-1)\Delta t), \ \frac{\mathrm{d}z}{\mathrm{d}t} = \begin{pmatrix} O & I \\ -I & O \end{pmatrix} \nabla H(z)$$

*with an analytic Hamiltonian* $H$. $\Delta t$ *is the sample time interval.*

**Assumption 4.3** *We assume that the matrices in the neural networks are uniformly bounded and hence the set of the neural networks is given as*

$\mathcal{H} = \{f_{\mathrm{NN}}(x) = \sigma_{\mathrm{enc}}(A_{\mathrm{enc}}x + b_{\mathrm{enc}}), \quad g_{\mathrm{NN}}(x) = \sigma_{\mathrm{dec}}(A_{\mathrm{dec}}x + b_{\mathrm{dec}}),$
$h_{\mathrm{NN}}(x) = \sigma_{\mathrm{symp}}(A_{\mathrm{symp}}x + b_{\mathrm{symp}}) \mid \|A_{\mathrm{symp}}\| < c_{\mathrm{enc}}, \quad \|A_{\mathrm{symp}}\| < c_{\mathrm{dec}}, \|A_{\mathrm{symp}}\| < c_{\mathrm{symp}}\}.$

*We also assume that the activation functions* $\sigma_{\mathrm{enc}}, \sigma_{\mathrm{dec}}$ *and* $\sigma_{\mathrm{symp}}$ *are analytic and Lipschitz continuous and the Lipschitz constants are bounded by* $\rho_{\mathrm{enc}}, \rho_{\mathrm{dec}}, \rho_{\mathrm{symp}}$, *respectively.*

**Assumption 4.4** *For normed spaces $X_1$, $X_2$, ..., $X_m$, we assume that the norm of the product space $X_1 \times X_2 \times \cdots \times X_m$ is defined by the sum of the norms: $\|\cdot\|_{X_1} + \cdots + \|\cdot\|_{X_m}$.*

**Assumption 4.5** *The loss function $\mathcal{L}$ can be regarded as a function that maps*

$$\left( f_{\mathrm{NN}}(x_j^{(1)}), \ldots, f_{\mathrm{NN}}(x_j^{(m_{\mathrm{step}})}), \, g_{\mathrm{NN}}(x_j^{(1)}), \ldots, \, g_{\mathrm{NN}}(x_j^{(m_{\mathrm{step}})}), h_{\mathrm{NN}}(x_j^{(1)}), \ldots, h_{\mathrm{NN}}(x_j^{(m_{\mathrm{step}})}), \right.$$

$$\left. (x_j^{(1)}, \ldots, x_j^{(m_{\mathrm{step}})}) \right)$$

*to a real number for each jth orbit. We assume this function is $\rho_{\mathcal{L}}$-Lipschitz continuous.*

The following are the main results of this paper.

**Theorem 4.6** *Under the above Assumptions, the covering number of the model is bounded by*

$$N(r/\rho_{\mathcal{L}}(c_{\mathrm{dec}}\rho_{\mathrm{dec}} + c_{\mathrm{symp}}\rho_{\mathrm{symp}} + 1)(c_{\mathrm{dec}}\rho_{\mathrm{dec}} + c_{\mathrm{symp}}\rho_{\mathrm{symp}} + 1)(m_{\mathrm{step}} \max\{1, \rho_{\psi}^{m_{\mathrm{step}}}\})),$$
$$B^{m_{\mathrm{orbit}}}).$$

For the proof of this theorem, see Appendix A.1. Since $B^{m_{\mathrm{orbit}}}$ is a compact set in a Euclid space from Assumption 4.1, the estimation of its covering number has been established. For example, its covering number in the 2-norm is estimated by using Lemma 3.6. The covering number in $m$-norms for $m$'s other than 2 can be estimated in the same way by using the equivalence of the norms in finite-dimensional spaces. Then, the application of Lemma 3.3 and Lemma 3.5 gives a generalization error bound. If the 2-norm is employed, the Rademacher complexity is shown to be $O(1/\sqrt{m_{\mathrm{orbit}}})$ by using Lemma 3.6. Note that Rademacher complexity mainly depends on the number of trajectories, not on the total number of the data. In fact, in the problem set-up of this study, only the initial values of the trajectories are chosen at random. Therefore, the generalization error depends on the number of the initial values of the trajectories, which means that for the models to be accurate, it is not enough to simply have a large number of data; there should be a large number of randomly sampled trajectory data.

Note also that the estimation of this theorem holds not only for Hamiltonian systems, but also for general physical systems, such as those involving friction. In particular, the physical model does not need to define a symplectic map. In addition, this theorem also holds for data with small time intervals.

**Numerical Experiments**  We performed an experiment to see how the number of trajectories affects the efficiency of learning. The experiments are identical to the 3body problem shown in the original HNNs paper Greydanus et al. (2019). See Greydanus et al. (2019) for the details of the experiments. We used the officially released codes[1] of HNN for this experiment. Regarding the data in this code, the trajectory data are given. More precisely, given the number of time steps $m_{\mathrm{step}}$ and the number of trajectories $m_{\mathrm{orbit}}$, numerical solutions of the target differential equation from random initial conditions are computed numerically and $m_{\mathrm{step}}$ snapshots are sampled for each orbit. We only changed $m_{\mathrm{step}}$ and $m_{\mathrm{orbit}}$ and the other settings were unchanged. In the experiments of Greydanus et al. (2019), $m_{\mathrm{step}}$ and $m_{\mathrm{orbit}}$ are set to $m_{\mathrm{step}} = 20$ and $m_{\mathrm{orbit}} = 5000$. In our experiments, we changed the number of orbits $m_{\mathrm{orbit}}$ to 5000, 2500, 1250 and 625. The number of time steps $m_{\mathrm{step}}$ is determined so that the number of the total data is unchanged.

The results are shown in Table 2. As expected, the accuracy of the neural network models becomes worse when the number $m_{\mathrm{orbit}}$ of the orbits becomes smaller; however, although, theoretically, it was expected that the generalization error would be proportional to $1/\sqrt{m_{\mathrm{orbit}}}$, in practice, when the number of orbits is not large enough, the training error did not decrease sufficiently. This should be because reducing the number of orbits increases the time steps, which makes the time period required to predict each orbit longer and the prediction task more difficult; however, this must be investigated in more detail in future work.

## 4.2 HAMILTONIAN INTERPOLATION

Second, we investigate whether the interpolated images follow a Hamilton equation or not. An important application of deep physical models is the interpolation of images from videos while

---

[1]https://github.com/greydanus/hamiltonian-nn

Table 2: Training results of Hamiltonian neural networks for the 3-body problem. Although the total number of data is constant, the accuracy improves as the number of trajectories increases.

| $m_{\text{orbit}}$ | $m_{\text{step}}$ | TRAINING ERROR | TEST ERROR | DIFFERENCE |
|---|---|---|---|---|
| 625 | 160 | 8.3945E+01 ± 1.4453E+02 | 2.3640E+01 ± 1.5563E+01 | 6.0305E+01 |
| 1250 | 80 | 3.9873E+00 ± 2.8671E+00 | 9.9912E+00 ± 1.5279E+00 | 6.0039E+00 |
| 2500 | 40 | 1.6950E+00 ± 1.4612E+00 | 2.2142E+00 ± 1.5869E+00 | 0.5192E+00 |
| 5000 | 20 | 8.5064E-02 ± 3.0044E-02 | 4.6424E-01 ± 4.2729E-01 | 3.7918E-01 |

preserving the laws of physics. As an application of the error analysis, we show that it is certainly possible to interpolate the images from videos so that the laws of physics are preserved, provided that the generalization error is sufficiently small.

**Assumption 4.7** *We assume that $h_{\text{NN}}$ is symplectic.*

**Assumption 4.8** *For the loss function $l_2$, we assume that the following holds: if the expectation value of $l_2$ is sufficiently small, then the map determined by $h_{\text{NN}}$ is close enough to the time $\Delta t$ map $\psi_{\Delta t}$ of the Hamilton equation satisfied by the latent variable $z$. More precisely, we assume that $\psi_{\Delta t}^{-1}$ exists, and that there exists a constant $c_{l_2}$ such that the operator norm of the difference between the two operators is bounded by*

$$\|h_{\text{NN}} \circ \psi_{\Delta t}^{-1} - \text{Id}\| \leq c_{l_2} E[\mathcal{L}_2].$$

For Assumption 4.8, for example, a sufficiently high order Sobolev norm can be used as the loss function. In fact, if the order is sufficiently large than $n_h$, the Sobolev embedding theorem gives a bound of the error in $L^\infty$ norm. Thus, since the error is uniformly bounded in the domain of $h_{\text{NN}}$, the map defined by $h_{\text{NN}}$ is close enough to the time $\Delta t$ map of the Hamilton equation satisfied by the latent variable $z$. Besides, the existence of $\psi_{\Delta t}^{-1}$ is not a strong assumption because Hamiltonian systems are typically inversible. On the other hand, note that we do not place the assumption that the time interval of the images to be interpolated is sufficiently small. Rather, we suppose that the time intervals may be quite large.

**Theorem 4.9** *Suppose that the loss function $\mathcal{L}$ for the training data satisfies $\mathcal{L} \leq \varepsilon_1$. Then, under the above Assumptions, for arbitrary $\varepsilon_2 > 0$, there exists $0 < \delta < 1$ such that with probability $1 - \delta$ there is an Hamiltonian $\hat{H} : \mathbb{R}^{n_h} \to \mathbb{R}$ that defines Hamiltonian system of which the time-1 map is an $O(c_{l_2}(\varepsilon_1 + 2\mathcal{R}_{m_{\text{orbit}}}) + \varepsilon_2)$ approximation to the symplectic map $h_{\text{NN}}$.*

For the proof, see Appendix A.2. Roughly, as seen in Figure 1, first, the map $h_{\text{NN}}$ is decomposed into a composition of two maps that are related to Hamiltonian systems, and then the splitting method is applied. Theorem 4.9 means that with a certain probability the trained model admits a Hamiltonian interpolation which is close to the true Hamiltonian system.

## 5 CONCLUSION

Recently, neural network methods for learning the equations of motion from observational data have attracted much attention. Although the theoretical analysis of such models has been gradually developed, a generalization error for such models trained with trajectory data has not yet been investigated. To the best of our knowledge, this paper is the first work on the error analysis for such a case. Our analysis shows an error bound that is dependent on the number of trajectories, not on the number of the whole data.

A possible application of these models is to interpolate images while preserving the laws of physics. Such interpolation is called Hamiltonian interpolation. In this paper, we also consider this application and show that under certain assumptions Hamiltonian interpolation is certainly possible by combining the autoencoder and a deep physical model that defines a symplectic map.

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
