## APPENDIX: PROOFS OF THE THEOREMS

### A.1 PROOF OF THE GENERALIZATION ERROR BOUND

In this section, we prove Theorem 4.6, i.e., the generalization error bound. To this end, the covering number of the set of the values of the loss functions must be estimated. Although the data set is given as

$$(x_1^{(1)}, \ldots, x_1^{(m_{\text{step}})}, x_2^{(1)}, \ldots, x_{m_{\text{orbit}}}^{(m_{\text{step}})}) \subset B^{m_{\text{step}} \times m_{\text{orbit}}},$$

from Assumption 4.1, these data are obtained by repeatedly applying the map $\psi$ to the initial data $(x_1^{(1)}, \ldots, x_{m_{\text{orbit}}}^{(1)}) \subset B^{m_{\text{orbit}}}$. This operation induces a map $\mathcal{O} : B^{m_{\text{orbit}}} \to B^{m_{\text{step}} \times m_{\text{orbit}}}$. Using the Lipschitz constant of the map $\psi$, for each $(x_1^{(1)}, \ldots, x_1^{(m_{\text{step}})}, x_2^{(1)}, \ldots, x_{m_{\text{orbit}}}^{(m_{\text{step}})}), (\tilde{x}_1^{(1)}, \ldots, \tilde{x}_1^{(m_{\text{step}})}, \tilde{x}_2^{(1)}, \ldots, \tilde{x}_{m_{\text{orbit}}}^{(m_{\text{step}})})$, we have

$$\begin{aligned}
&\|(x_1^{(1)}, \ldots, \ldots, x_{m_{\text{orbit}}}^{(m_{\text{step}})}) - (\tilde{x}_1^{(1)}, \ldots, \tilde{x}_{m_{\text{orbit}}}^{(m_{\text{step}})})\| \\
&= \|(x_1^{(1)}, \ldots, \psi^{m_{\text{step}}}(x_1^{(1)}), x_2^{(1)}, \ldots, \psi^{(m_{\text{step}})}(x_{m_{\text{orbit}}}^{(1)})) - (\tilde{x}_1^{(1)}, \ldots, \psi^{(m_{\text{step}})}\tilde{x}_{m_{\text{orbit}}}^{(1)}))\| \\
&= \|x_1^{(1)} - \tilde{x}_1^{(1)}\| + \cdots + \|\psi^{m_{\text{step}}}(x_1^{(1)}) - \psi^{m_{\text{step}}}(\tilde{x}_1^{(1)})\| \\
&\quad + \cdots + \|\psi^{(m_{\text{step}})}(x_{m_{\text{orbit}}}^{(1)}) - \psi^{(m_{\text{step}})}(x_{m_{\text{orbit}}}^{(1)})\| \\
&\leq m_{\text{step}} \max\{1, \rho_\psi^{m_{\text{step}}}\}(\|x_1^{(1)} - \tilde{x}_1^{(1)}\| + \cdots + \|x_{m_{\text{orbit}}}^{(1)} - \tilde{x}_{m_{\text{orbit}}}^{(1)}\|) \\
&= m_{\text{step}} \max\{1, \rho_\psi^{m_{\text{step}}}\}\|(x_1^{(1)} - \tilde{x}_1^{(1)}, \ldots, x_{m_{\text{orbit}}}^{(1)} - \tilde{x}_{m_{\text{orbit}}}^{(1)})\|,
\end{aligned}$$

which shows the Lipschitz constant of $\mathcal{O}$ is bounded by $m_{\text{step}} \max\{1, \rho_\psi^{m_{\text{step}}}\}$. Hence, we obtain

$$N(r, \mathcal{O}(B^{m_{\text{orbit}}})) \leq N(\frac{r}{m_{\text{step}} \max\{1, \rho_\psi^{m_{\text{step}}}\}}, B^{m_{\text{orbit}}}).$$

To compute the loss functions $\mathcal{L}_1$ and $\mathcal{L}_2$, these extended data are first input into $f_{\text{NN}}$ to form the set of the latent variables:

$$Z := \{(f_{\text{NN}}(x), x) \mid x \in \mathcal{O}(B^{m_{\text{orbit}}})\} \subset B^{m_{\text{step}} \times m_{\text{orbit}}} \times B^{m_{\text{step}} \times m_{\text{orbit}}}.$$

For each $x, \tilde{x} \in \mathcal{O}(B^{m_{\text{orbit}}})$, we have

$$\|(f_{\text{NN}}(x), x) - (f_{\text{NN}}(\tilde{x}), \tilde{x})\| = \|f_{\text{NN}}(x) - f_{\text{NN}}(\tilde{x})\| + \|x - \tilde{x}\| \leq (c_{\text{enc}}\rho_{\text{enc}} + 1)\|x - \tilde{x}\|.$$

This estimate shows that

$$N(r, Z) \leq N\left(\frac{r}{(c_{\text{enc}}\rho_{\text{enc}} + 1)}, \mathcal{O}(B^{m_{\text{orbit}}})\right).$$

Second, the set $Z$ should be transformed to

$$\{(f_{\text{NN}}(x), g_{\text{NN}}(x), h_{\text{NN}}(x), x) \mid x \in \mathcal{O}(B^{m_{\text{orbit}}})\}.$$

Similarly to the estimation of the map from $\mathcal{O}(B^{m_{\text{orbit}}})$ to $Z$, we get

$$N(r, \{(f_{\text{NN}}(x), g_{\text{NN}}(x), h_{\text{NN}}(x), x) \mid x \in \mathcal{O}(B^{m_{\text{orbit}}})\}) \leq N(\frac{r}{c_{\text{dec}}\rho_{\text{dec}} + c_{\text{symp}}\rho_{\text{symp}} + 1}, Z).$$

Because we assume that the loss function $\mathcal{L}$ is $\rho_{\mathcal{L}}$-Lipschitz continuous, we have

$$\begin{aligned}
&N(r, \{\mathcal{L}(f_{\text{NN}}(x), g_{\text{NN}}(x), h_{\text{NN}}(x), x) \mid x \in \mathcal{O}(B^{m_{\text{orbit}}})\}) \\
&\leq N(\frac{r}{\rho_{\mathcal{L}}(c_{\text{dec}}\rho_{\text{dec}} + c_{\text{symp}}\rho_{\text{symp}} + 1)}), Z).
\end{aligned}$$

Combining all of the above results yields the following inequality:

$$\begin{aligned}
&N(r, \{\mathcal{L}(f_{\text{NN}}(x), g_{\text{NN}}(x), h_{\text{NN}}(x), x) \mid x \in \mathcal{O}(B^{m_{\text{orbit}}})\}) \\
&\leq N(\frac{r}{\rho_{\mathcal{L}}(c_{\text{dec}}\rho_{\text{dec}} + c_{\text{symp}}\rho_{\text{symp}} + 1)}), Z) \\
&\leq N(r/\rho_{\mathcal{L}}(c_{\text{dec}}\rho_{\text{dec}} + c_{\text{symp}}\rho_{\text{symp}} + 1)(c_{\text{dec}}\rho_{\text{dec}} + c_{\text{symp}}\rho_{\text{symp}} + 1)), \mathcal{O}(B^{m_{\text{orbit}}})) \\
&\leq N(r/\rho_{\mathcal{L}}(c_{\text{dec}}\rho_{\text{dec}} + c_{\text{symp}}\rho_{\text{symp}} + 1)(c_{\text{dec}}\rho_{\text{dec}} + c_{\text{symp}}\rho_{\text{symp}} + 1)(m_{\text{step}} \max\{1, \rho_\psi^{m_{\text{step}}}\})), \\
&\quad B^{m_{\text{orbit}}}).
\end{aligned}$$

This shows Theorem 4.6.

## A.2 PROOF OF THE HAMILTONIAN INTERPOLATION

Let $\varepsilon_2 > 0$ be arbitrarily chosen. Suppose that the loss function for the training data satisfies $\mathcal{L} \leq \varepsilon_1$. Then, from Lemma 3.3, with probability at least $1 - \delta$, it holds that

$$E[\mathcal{L}_2] \leq E[\mathcal{L}] \leq \varepsilon_1 + 2\mathcal{R}_{m_{\text{orbit}}}(\mathcal{G}) + 3c\sqrt{\frac{2\ln\frac{4}{\delta}}{m_{\text{orbit}}}}.$$

When this inequality holds, from Assumption 4.8, we have

$$\|h_{\text{NN}} \circ \psi_{\Delta t}^{-1} - \text{Id}\| \leq c_{l_2} E[\mathcal{L}_2]$$

$$\leq c_{l_2}(\varepsilon_1 + 2\mathcal{R}_{m_{\text{orbit}}} + 3c\sqrt{\frac{2\ln\frac{4}{\delta}}{m_{\text{orbit}}}}).$$

Thus, if we let $\hat{\delta}$ be

$$\hat{\delta} = 4\exp(-\frac{m_{\text{orbit}}\varepsilon^2}{18c^2}),$$

at least probability $1 - \hat{\delta}$, the following inequality holds:

$$\|h_{\text{NN}} \circ \psi_{\Delta t}^{-1} - \text{Id}\| \leq c_{l_2} E[\mathcal{L}_2] \leq c_{l_2}(\varepsilon_1 + 2\mathcal{R}_{m_{\text{orbit}}} + \varepsilon_2).$$

Then, from the above estimate, $h_{\text{NN}} \circ \psi_{\Delta t}^{-1}$ is close to the identity. Also, this is a symplectic map because both $h_{\text{NN}}$ and $\psi_{\Delta t}^{-1}$ are symplectic. Hence, from Theorem 3.1, there exists a Hamiltonian flow $\hat{h}_{\text{NN}}$ that appoximates $h_{\text{NN}} \circ \psi_{\Delta t}^{-1}$ within the error

$$O(c_{l_2}(\varepsilon_1 + 2\mathcal{R}_{m_{\text{orbit}}} + \varepsilon_2)\exp(-\frac{1}{c_{l_2}(\varepsilon_1 + 2\mathcal{R}_{m_{\text{orbit}}} + \varepsilon_2)})).$$

Because $h_{\text{NN}}$ is written as

$$h_{\text{NN}} = (h_{\text{NN}} \circ \psi_{\Delta t}^{-1}) \circ \psi_{\Delta t} \simeq \hat{h}_{\text{NN}} \circ \psi_{\Delta t},$$

$h_{\text{NN}}$ is approximated by the composition of the two Hamiltonian flows $\hat{h}_{\text{NN}}$ and $\psi_{\Delta t}$. The error analysis of the splitting method shows that there exists a Hamiltonian flow $\tilde{h}_{\text{NN}}$ that approximates $\hat{h}_{\text{NN}} \circ \psi_{\Delta t}$ within the error $O(\|\hat{h}_{\text{NN}}\|\|\psi_{\Delta t}\|)$. This $\tilde{h}_n n$ approximates $h_{\text{NN}}$, and the approximation error is estimated by

$$\|h_{\text{NN}} - \tilde{h}_n n\| \leq \|h_{\text{NN}} - \hat{h}_{\text{NN}} \circ \psi_{\Delta t}\| + \|\hat{h}_{\text{NN}} \circ \psi_{\Delta t} - \tilde{h}_n n\|$$

$$= O(c_{l_2}(\varepsilon_1 + 2\mathcal{R}_{m_{\text{orbit}}} + \varepsilon_2)\exp(-\frac{1}{c_{l_2}(\varepsilon_1 + 2\mathcal{R}_{m_{\text{orbit}}} + \varepsilon_2)})) + O(\|\hat{h}_{\text{NN}}\|\|\psi_{\Delta t}\|)$$

Since $\|\hat{h}_{\text{NN}}\|$ is $O(c_{l_2}(\varepsilon_1 + 2\mathcal{R}_{m_{\text{orbit}}} + \varepsilon_2))$, the approximation error is estimated by $O(c_{l_2}(\varepsilon_1 + 2\mathcal{R}_{m_{\text{orbit}}} + \varepsilon_2))$. This completes the proof.