# OpenReview forum: "Generalization Error Analysis of Deep Physical Models With Latent Variables Trained on Trajectory Data"
_ICLR.cc/2024/Conference — ICLR 2024 Conference Withdrawn Submission_

### Official Review · Reviewer_pXTH · 2023-10-15

**Soundness:** 3 good
**Presentation:** 2 fair
**Contribution:** 2 fair
**Rating:** 3
**Confidence:** 4

**Summary:**

The authors consider the generalization properties of Hamiltonian Neural Networks when trained on trajectory data. Their main object of interest is an upper bound on the generalization gap which they obtain using Radmacher complexity-based bounds. Specifically, assuming the Neural Networks involved are p-Lifchitz, they obtain bounds on the covering number and use the latter to bound the complexity. In an additional part of the paper, they relate their results on the generalization gap to bound the deviation from "symplectic-ness" of the learned Hamiltonian.

**Strengths:**

1. The paper applies learning theory in a relatively fresh domain, that of HNNs with autoencoders.

2. The paper departs from the more standard case of random training data and attempts to tackle series data with random initial conditions.

3. It produces solid results, which may be a good addition to the technical literature on the topic.

**Weaknesses:**

I believe the authors' results merit publication in a specialized journal rather than in ICLR. The main reasons are the following
1. The authors do not give any compelling numerical evidence that their bound is tight or even "log-tight".
2. The authors' derivation falls into classical learning theory-based bounds, which, to the best of my knowledge, does not yield realistic bounds, unless Bayesian considerations are taken into account (e.g. Bayesian-PAC based bounds).
3. Even if one maintains that VC-dimension-style learning theory is an important part of the theory of deep learning, my hunch would be that the current work does not contain sufficient mathematical interest to be published in ICLR.

My more minor comments are that
1. The introduction is very wordy and contains many repetitions of similar statements.
2. I found what I believe are various math typos, for instance around Lemma 3.5. I think n and m are used interchangeably. Furthermore calligraphic R with an n sub-script and regular R. Similarly, capital and non-capital l are mixed in assumption 4.8. Runaway subscripts also appear many times in Appendix A2.

**Questions:**

1. Can the authors provide experiments on some more toy models (having say only 2-layer dense networks in each module) and compare their bound with experiments as one varies the number of trajectories or their sampling density?
2. Can the authors provide estimates of how well their bound works for the 3-body problem they studied numerically?
3. Can the authors draw some potential insights on architecture design based on their bound?

---

### Official Review · Reviewer_dY6S · 2023-10-31

**Soundness:** 3 good
**Presentation:** 3 good
**Contribution:** 3 good
**Rating:** 5
**Confidence:** 1

**Summary:**

This paper investigates the generalization error of a neural network model that learns for data of physical phenomena, particularly for trajectory data. The generalization error bound of a model for such data is provided with theoretical proof. The analysis claims that the number of trajectories is an important factor for better generalization rather than the number of data. The experiment assures that, for the three-body problem, the analysis is correct.

**Strengths:**

As the authors claim, as well as to the best of my knowledge, this is the first theoretical analysis of the generalization error for a neural network model for the trajectory data of a physics problem. The analysis provides the audience with an intuitive conclusion that highlights the importance of the number of trajectories for such tasks. Based on the analysis, the authors claim a possibility of Hamiltonian interpolation for images from a video containing physical phenomena while preserving the laws of physics such as the conservation of energy.

**Weaknesses:**

I find that the experiment could have been more thoroughly conducted and analyzed to validate the proposed theoretical observations. Please see below for more details.

**Questions:**

First of all, only one single example of the 3-body problem does not seem to be sufficient to validate the analysis strongly enough. I am curious whether this can be expected too when applying to other trajectory data with different physics problems.

I find the generalization error somewhat ambiguous. For instance, in the 3-body numerical experiment, I am wondering in what context the generalization error was analyzed. Is the experiment investigating the generalization for temporal extrapolation of trajectory? Is it investigating the generalization for unseen trajectories (e.g., interpolation or extrapolation of the initial conditions)?

In conjunction with the previous question, how much the variance of initial conditions will affect the generalization? I guess the data generated with different distributions of the initial conditions may affect the error.

It would be stronger if the work presented an actual experiment for the Hamiltonian interpolation.

---

### Official Review · Reviewer_QwHa · 2023-11-01

**Soundness:** 2 fair
**Presentation:** 2 fair
**Contribution:** 2 fair
**Rating:** 5
**Confidence:** 3

**Summary:**

This work analyzes the generalization error of deep physical models under the setting of random paths instead of complete randomness. The derived upper bound only depends on the number of trajectories, independent of the path length, meaning that for the model to be accurate, it is not enough to simply collect more data, but more (randomly sampled) trajectory data are preferred.
This insight is numerically verified on the 3-body problem.
As an important application, if one can guarantee a sufficiently small generalization error, the Hamiltonian neural network can interpolate images from videos while preserving the laws of physics (i.e. satisfying the Hamiltonian dynamics) even for a large frame (time) interval.

**Strengths:**

Highlights:

- This work steps further to a theoretical analysis of deep physical models trained with *trajectory* data.

- This work also theoretically proves the important Hamiltonian interpolation problem, particularly for image sequences with large frame intervals, by a *combined* study of three fields: statistical learning theory (generalization gap analysis), symplectic geometry (Hamiltonian approximation of near-identity symplectic maps), and numerical analysis (splitting integrator).

**Weaknesses:**

Concerns:

- The proof of generalization gap is quite straightforward. The analysis is based on the classical tool (covering number) in statistical learning theory, but the sequence is handled only via a repeated use of Lipschitz transformations. This argument holds in general, but is without further *dynamical* information and hence not away from the complete randomness setting.

- The tightness of bounds: It is known that the Rademacher complexity can be upper bounded by the covering number, but this work performs a covering number analysis.

- Simulations: There is only one experiment on the 3-body problem.

**Questions:**

Corresponding with concerns proposed in the **Weaknesses** section, I have the following questions:

-  Can any dynamical conditions or physical laws on sequences be considered? Insightful examples are also welcomed to make it a real "trajectory" result.

- Is it possible to bound the Rademacher complexity to get a tighter upper bound? If possible, will this bound be related with both the number and length of paths? The experimental results (Table 2) only demonstrate that the path number has a superior effect over the path length, but the path length also matters in many practical applications.

- Besides the classical 3-body problem, can you numerically verify the superior effect of the path number over the path length on public datasets and benchmarks? Is it possible to numerically test the Hamiltonian interpolation part, particularly on the effect of varied generalization errors?

Other details:

- Assumption 4.1: For each trajectory, the transition map $\psi$ is $\rho_\psi$-Lipschitz continuous. In the meanwhile, all the data are uniformly bounded. So do we have $\rho_\psi \le 1$?

- Assumption 4.2: This work assumes a Hamiltonian latent variable. Can you kindly provide some examples for better readability?

- Theorem 4.6: Is it a typo to have "dec"-related constants for two times and without "enc"-related constants in the upper bound? It seems that the same typo occurs in the proof.

- Assumption 4.8: Can you kindly provide some examples of $h_{\text{NN}}$ and $\psi_{\Delta t}$ for better understanding?

- Proofs: In the last sentence of the proof of Hamiltonian interpolation, can you explain why $\||\hat{h}_{\text{NN}}\||$ is $O(c\_{l\_2}(\varepsilon_1+2\mathcal{R}\_{m\_{\text{orbit}}}+\varepsilon_2))$?

    - It seems that $\hat{h}_{\text{NN}} \approx h\_{\text{NN}} \circ \psi\_{\Delta t}^{-1}$ and $\||h\_{\text{NN}} \circ \psi\_{\Delta t}^{-1} - \mathrm{Id}\||$ is $O(c\_{l\_2}(\varepsilon_1+2\mathcal{R}\_{m\_{\text{orbit}}}+\varepsilon_2))$.

---

### Official Review · Reviewer_2r94 · 2023-11-06

**Soundness:** 4 excellent
**Presentation:** 3 good
**Contribution:** 4 excellent
**Rating:** 8
**Confidence:** 1

**Summary:**

This manuscript evaluates the generalization performance of a deep physics model, a neural network model that learns equations of motion from observed data. The theoretical results obtained show that to improve generalization performance, it is important to collect data from many trajectories rather than simply collecting a large amount of data. The paper also shows that deep physics models can interpolate images from video images while preserving physical laws such as conserved quantities if the generalization error is sufficiently small.

**Strengths:**

This manuscript is a significant study in the evaluation of the generalization performance of deep physics models, which has been lacking in the studies of these models, and is considered to be a significant topic. It is also highly significant that the paper provides not only the evaluation of generalization performance but also important findings for applications, such as the usefulness of collecting data from a large number of trajectories to improve the generalization performance. It is also significant that the authors have enhanced the validity of their assertions by confirming their theoretical findings with experimental results.

**Weaknesses:**

To further elaborate on the theoretical findings, the paper might benefit from further validation by including validation in a more diverse and broader range of physical systems.

**Questions:**

To further elaborate on the theoretical findings, the paper might benefit from further validation by including validation in a more diverse and broader range of physical systems.